# Electromyography of Extrinsic and Intrinsic Ear Muscles in Healthy Probands and Patients with Unilateral Postparalytic Facial Synkinesis

**DOI:** 10.3390/diagnostics12010121

**Published:** 2022-01-05

**Authors:** Hanna Rüschenschmidt, Gerd Fabian Volk, Christoph Anders, Orlando Guntinas-Lichius

**Affiliations:** 1Department of Otorhinolaryngology, Jena University Hospital, 07747 Jena, Germany; hanna.rueschenschmidt@web.de (H.R.); fabian.volk@med.uni-jena.de (G.F.V.); 2Facial Nerve Center, Jena University Hospital, 07747 Jena, Germany; 3Center for Rare Diseases, Jena University Hospital, 07747 Jena, Germany; 4Division for Motor Research, Pathophysiology and Biomechanics, Department for Trauma-, Hand- and Reconstructive Surgery, Jena University Hospital, 07743 Jena, Germany; christoph.anders@med.uni-jena.de

**Keywords:** auricular muscles, facial muscles, human, facial palsy, electrophysiology, ear wiggling

## Abstract

There are currently no data on the electromyography (EMG) of all intrinsic and extrinsic ear muscles. The aim of this work was to develop a standardized protocol for a reliable surface EMG examination of all nine ear muscles in twelve healthy participants. The protocol was then applied in seven patients with unilateral postparalytic facial synkinesis. Based on anatomic preparations of all ear muscles on two cadavers, hot spots for the needle EMG of each individual muscle were defined. Needle and surface EMG were performed in one healthy participant; facial movements could be defined for the reliable activation of individual ear muscles’ surface EMG. In healthy participants, most tasks led to the activation of several ear muscles without any side difference. The greatest EMG activity was seen when smiling. Ipsilateral and contralateral gaze were the only movements resulting in very distinct activation of the transversus auriculae and obliquus auriculae muscles. In patients with facial synkinesis, ear muscles’ EMG activation was stronger on the postparalytic compared to the contralateral side for most tasks. Additionally, synkinetic activation was verifiable in the ear muscles. The surface EMG of all ear muscles is reliably feasible during distinct facial tasks, and ear muscle EMG enriches facial electrodiagnostics.

## 1. Introduction

The human auricle contains three extrinsic and six intrinsic muscles [1]. These muscles have been considered vestigial in humans, and are rarely under voluntary control [2]. Like the mimic muscles, all ear muscles are innervated by the facial nerve [1]. Needle and surface electromyography (EMG) are part of routine electrodiagnostics in patients with facial nerve dysfunction, mainly for facial palsy [3]. In these patients, EMG is routinely performed from the mimic muscles. EMG data on human ear muscles are sparse. Berzin and Fortinguerra published data for the three external muscles (anterior, superior and posterior auricular) of 30 healthy men [4]. They showed EMG activity in all these muscles but never isolated activity in only one muscle during smiling and yawning. Serra et al. analyzed the posterior auricular muscle in patients with acute facial paralysis [5]. Due to the acute paralysis, EMG activity disappears in this ear muscle like in mimic muscles. Posterior auricular muscle EMG activity preceded the recovery of EMG signs in other facial muscles by 10–30 days. Furthermore, they revealed in patients with postparalytic facial synkinesis synchronous EMG activity with the orbicularis oculi muscle during eye blink. This is a sign that the ear muscles are also involved in the process of postparalytic synkinesis. Recently, such postparalytic facial-auricular synkinesis was also reported in a case report [6]. The ear muscles are not only of interest in patients with facial palsy. The ear muscles are part of a complex neural brainstem network for the postauricular reflex [7]. Therefore, the measurement of ear muscle EMG activity offers so-far unexploited potential for the monitoring of emotional states, brainstem lesion diagnostics or stroke manifestations [2]. Finally, the ear muscles are beneficial for individuals with quadriplegia because these muscles are usually not affected by high-level spinal lesions [2]. For instance, such patients can learn to steer a wheelchair with ear muscle activation [8]. This also opens the field for other neuroprosthetic applications based on auricular muscles.

Therefore, we thought that it would be worthwhile to systematically investigate: (1) the exact localization of all extrinsic and intrinsic ear muscles in cadaver preparations for optimal placement of the EMG electrodes, (2) the EMG activity in all extrinsic and intrinsic ear muscles in healthy individuals and (3) the changes of EMG activity in patients with postparalytic facial synkinesis.

## 2. Material and Methods

This study was approved by the Ethics Committee of Jena University Hospital (No. 2018-1103-BO). All participants provided written informed consent. Twelve healthy probands (8 female, 4 male; age: 18–29 years) were examined by one-channel and multi-channel EMG. Additionally, needle EMG was performed in one healthy proband only. Surface EMG was performed in all 12 probands. Seven patients (4 female, 3 male; age: 18–60 years) with postparalytic facial synkinesis after acute unilateral (5 right side, 2 left side) peripheral facial paralysis (mean time since onset: 34.7 months) were examined. The patients received surface one-channel EMG. All measurements were performed on both sides of the face. Additionally, the anatomical preparations were performed on two hemifaces.

### 2.1. Cadaver Preparations

First, cadaver preparations were performed to better understand the exact localization and size of the ear muscles. The anatomical preparations were carried out on two hemifaces from body donors at the forensic medicine department of the University Hospital Hamburg-Eppendorf, Germany. First, the pre-, supra- and postauricular cutis and subcutis were dissected. The scalp was detached from ventral to dorsal around the cartilaginous auricle. As the auricular muscles are not different to other facial muscles, they were not covered by fascia. After removing the subcutaneous fat tissue, the three extrinsic ear muscles could be visualized. For the preparation of the ventral side of the auricle, the cutis was dissected away from the cartilage and the cartilaginous tragus exposed. For further dissection of the tragicus muscle, the cartilage was slit on the dorsal posterior margin of the tragus so that the cartilage could be unfolded. The helicis major and helicis minor muscles were exposed on the crus helicis. Following the course of the helix, the skin was detached down to the earlobe and the antitragicus muscle was exposed. Finally, the transversus auriculae and the obliquus auriculae muscle were dissected on the back side after removal of the cutis.

The photographic documentation of the prepared ear muscles was compared with anatomical drawings from anatomy atlases [1,9,10,11,12]. Both were used as a guide for the positioning of the fine needle electrodes and surface electrodes. Graphics were created which illustrate the size and position of the ear muscles and on which the exact positioning of the electrodes could be displayed.

### 2.2. One-Channel Needle and Surface Electromyography Setting

Needle EMGs were first performed in one healthy proband using optimal recording spots based on the cadaver dissections. The region of optimal signaling was thereafter used for the surface EMG recordings. A standard EMG system was used (Medelec Synergy T5, VIASYS Healthcare, Höchberg, Germany; high-pass filter: 20 Hz; low-pass filter: 5 kHz). For needle EMG, concentric electrodes were used (Neuroline concentric needle, 50 × 0.45 mm; Ambu, Bad Nauheim, Germany). For surface EMG, surface electrodes were applied (Neuroline 720 Neurology Surface Electrodes 72015- K/10, Ambu, Bad Nauheim, Germany).

All EMG examinations were performed in supine position. The auricle and surrounding skin were disinfected and degreased with a skin disinfectant. The reference electrode was positioned on the spinous process of the seventh cervical vertebra. Only before performing the needle EMGs were the puncture sites anesthetized with an anesthetic cream (EMLA; 25 mg/g lidocaine plus 25 mg/g prilocaine, Aspen, Munich, Germany). In healthy probands, the examinations started on the left side, followed by the right side. In patients, first the contralateral side and then the affected sided was analyzed. The sequence of recordings was always the same: auricularis anterior, auricularis superior, auricularis posterior, tragicus, antitragicus, helicis major, helicis minor, transversus auriculae and finally obliquus auriculae muscle. The probands had to perform the following mimic exercises, always in the same sequence: smiling, pursing lips, nose wrinkling, frowning, drawing eyebrows together, ipsilateral gaze, contralateral gaze and finally ear wiggling. Each task was performed 48 times. The movements were demonstrated by the examiner and practiced together with the examiner before starting the recordings. For quality control, all EMG recordings were video-documented (Appendix A). The amplitude of the EMG signal from the recorded ear muscle during the specific exercise was evaluated off-line using the video recordings and the following standard 4-point classification system [3]: no increase compared to baseline activity in resting state (=0), slight increase (=1), moderate increase (=2) and strong increase (=3). Examples for the classification are shown in Appendix A.

### 2.3. Multi-Channel Surface Electromyography Setting

For multi-channel EMG, a standard EMG system (Tower of Measurement, DeMeTec, Langgöns, Germany) and amplifier (Biovision, Wehrheim, Germany; sampling rate: 4096 Hz; amplitude resolution: 60 nV/bit) were used. Three ear muscles (auricularis superior, auricularis posterior, tragicus), a mimic muscle (risorius) and two chewing muscles (temporalis, masseter) were synchronously recorded. The recordings were performed in a sitting position. Surface electrodes (Kendall EGC electrodes H93SG, 42 mm × 24 mm; Covidien, Dublin, Ireland) were applied (Appendix A). These electrodes have the advantage that they can be trimmed. By trimming, electrode overlap can be avoided. The probands had to perform the following mimic exercises, always in the same sequence as described for the single-channel recording (see above). In addition, the following tasks were performed: showing teeth, clenching teeth, chewing on the ipsilateral side and chewing on the contralateral side. All EMG recordings were video-documented. For classification of the EMG activity, the same classification systems as for the single-channel analysis were used (see above).

### 2.4. Statistical Analysis

Statistical analyses were performed using Microsoft Excel (V.2016, Microsoft Corporation, Seattle, WA, USA) and using IBM SPSS v.26.0 statistical software for Windows (Chicago, IL, USA). If not otherwise reported, mean values ± standard deviations of the EMG recordings of the healthy probands or the patients are presented. Differences between two dependent subgroups (right versus left side in healthy probands or postparalytic side versus contralateral side) for ordinal EMG classification data were compared with the Wilcoxon test. For all statistical tests, significance was two-sided and set to *p* < 0.05.

## 3. Results

### 3.1. Cadaver Preparations of the Extrinsic and Intrinsic Ear Muscles

Appendix A summarizes the preparations of the ear muscles. Nearly all muscles could be detected. The helicis minor muscle could not be clearly depicted in both cadavers. The tragicus and the antitragicus were located within the tragal and antitragal cartilage, respectively. The auricularis posterior muscle had two muscle bellies in one cadaver. The transversus auriculae muscle was clearly visible in only one preparation. Optimal spots for the surface EMG recordings of all ear muscles (Figure 1) were defined based on the cadaver preparations and needle EMG results (see below).

### 3.2. One-Channel Needle Electromyography of the Extrinsic and Intrinsic Ear Muscles in One Healthy Proband

Except for the transversus auriculae muscle on the left side and the obliquus auriculae muscle on the right side, needle EMG was feasible for all extrinsic and intrinsic ear muscles (Appendix A). The optimal hot spots for the needle EMG of individual ear muscles are shown in Appendix A. Overall, the external muscles showed higher activity than the intrinsic muscles. The strongest activity was seen in all muscles during ear wiggling and smiling. The obliquus auriculae showed the weakest activity. The transversus auriculae was only activated during lateral gaze. Frowning only activated the extrinsic ear muscles, whereas the same muscles showed no activity during drawing of the eyebrows together. The activity pattern during needle EMG was not significantly different from surface EMG in the one individual receiving both needle and surface EMG analyses (data of the comparison not shown).

### 3.3. One-Channel Surface Electromyography of the Extrinsic and Intrinsic Ear Muscles in Healthy Probands

Surface EMG could be recorded from all ear muscles in all healthy subjects. There was no significant right–left side difference (Table 1). In general, the ear muscle EMG activity was highest during smiling and ear wiggling. The lowest activity was seen during ipsilateral or contralateral gaze. Only the obliquus auriculae and the transversus auriculae muscle showed a robust activity during the isolated lateral gaze.

### 3.4. Multi-Channel Surface Electromyography of the Extrinsic and Intrinsic Ear Muscles in Healthy Probands

The mimic exercises (smiling, pursing lips, nose wrinkling, frowning, drawing eyebrows together, showing teeth) activated the risorius as a mimic muscle, but also synchronously at least two ear muscles and one or both chewing muscles (Table 2). Ear wiggling showed the strongest co-activation of all recorded muscles. The co-activation during the lateral gaze was weaker but clearly visible in mimic and ear muscles, but very weak in the temporalis muscle. The masseter showed no activity during the lateral gaze movements. The chewing tasks (clenching teeth, chewing ipsilateral, chewing contralateral) triggered a strong activation in all muscles except for the auricularis posterior muscles. This muscle remained silent during chewing tasks.

### 3.5. One-Channel Surface Electromyography of the Extrinsic and Intrinsic Ear Muscles in Patients with Postparalytic Facial Synkinesis

Like in healthy individuals, the strongest EMG activity in ear muscles was seen during smiling on both sides, and for ear wiggling on the contralateral side (Table 3). Interestingly, the activity on the postparalytic side was much lower during ear wiggling. Especially, nose wrinkling showed a much higher activity on the postparalytic side than in normal individuals. The average EMG activity was always higher on the postparalytic side than on the contralateral side. An example for the anterior and posterior muscles is shown in Figure 2. Even in this small sample size, the difference was significantly different (*p* < 0.05) for several muscles and tasks. EMG activity was also seen in other ear muscles than the obliquus auriculae and the transversus auriculae muscle.

## 4. Discussion

With this work, data are available for the first time on the implementation of EMGs on all intrinsic and extrinsic ear muscles. The co-activation of the ear muscles when smiling, pursing the lips, wrinkling the nose and making other facial movements could be shown by means of surface and needle EMG recordings. The results of the needle EMG with its very high spatial resolution and a measuring field of only approximately one cubic millimeter were important to define the hot spots for the recording and to confirm that the measured activity actually came from the inserted ear muscles of interest and not from the much larger neighboring muscles (e.g., the masseter muscles). Surface EMGs of all extrinsic and intrinsic ear muscles were feasible and meaningful, since individual activity patterns could be determined during co-activation through facial movements. Synchronous activation of ear muscles was the rule, and isolated activation was the exception. Hence, the results of Berzin and Fortinguerra could be confirmed for all ear muscles [4].

The postauricular muscle reflex activates the posterior auricular muscle in response to short and abrupt sounds [13]. This brainstem reflex is of interest because it is preserved in case of facial nerve lesion distal to the stylomastoid foramen but disturbed after more proximal lesions [14]. This reflex was not measured in the current study, but should be included in future investigations. However, the oculo-auricular phenomenon (i.e., the co-activation of ear muscles during lateral gaze) was investigated [15]. In concordance with previous investigations, we could show that especially ipsilateral gaze led to a strong (sometimes isolated) activation of the transversus auriculae and obliquus auriculae muscles [15]. Furthermore, it could be confirmed that lateral gaze enhances postauricular muscle reflex [16].

Needle electromyography is the standard method to objectively confirm facial synkinesis—that is, the abnormal involuntary facial movement that occurs with voluntary movement of a different facial muscle group [3]. Surface EMG also makes it possible to prove facial synkinesis [17]. The present study is the first to systematically show that postparalytic synkinesis also includes the extrinsic and intrinsic ear muscles. We showed complex patterns of involuntary ear muscle activity during mimic tasks beyond the typical synkinesis patterns seen in the mimic muscles of the patients. In most cases the EMG activity was heavily increased in comparison to the activity normally seen in these small muscles. One case report was published in 2020 showing oculo-auricular or oro-auricular synkinesis during similar mimic tasks, as in the present study [6]. Analysis of the ear muscles should be included into the repertoire of facial-muscle EMG analysis for patients with facial nerve diseases. Most ear muscles are innervated by the posterior auricular nerve [18]. Only the superior and anterior auricular muscles are innervated by the temporal branches of the facial nerve. The posterior auricular nerve arises from the facial nerve close to the stylomastoid foramen. Therefore, ear muscle EMG should be helpful to differentiate between intratemporal and extratemporal facial nerve trauma.

The study had several limitations. A standard examination procedure could be established and the ear muscle EMGs were highly reproducible, but overall the sample sizes of healthy volunteers and patients with postparalytic facial synkinesis were low. During most tasks, the average EMG activation was higher on the postparalytic than on the contralateral side. Due to the small samples, the difference reached a statistically significant difference only for some of these results. We expect clearer results when increasing the sample sizes in the future. The qualitative score to assess the changes in EMG activity was derived from clinical EMG scores used for the classification of facial muscle EMG [3]. Schmidt and Thoden use a similar score for classifying ear muscle activity [19]. To check the reproducibility of the evaluation of the EMGs by means of the qualitative score, an evaluation of the EMG activity changes by different examiners will be necessary in future trials. Even better, an automated EMG analysis should be developed. Recently, such a system was developed for the automated analysis of multi-channel mimic muscle EMG recordings [20].

Another recent study demonstrated that the ability to activate the posterior auricular muscle can be learned and used intuitively to steer a wheelchair with a myoelectric auricular control system [8]. The present study also shows that the other ear muscles show sufficient and reproducible activity to be used by assistive technologies even for more complex steering or other tasks.

## 5. Conclusions

Surface EMG recordings of all intrinsic and extrinsic ear muscles were feasible and reproducible. Distinct mimic movement tasks in which the ear muscles are active were defined. Mostly, several ear muscles were activated. Furthermore, in patients with postparalytic facial nerve syndrome, synkinetic activation of the ear muscles is the rule.

## Figures and Tables

**Figure 1 diagnostics-12-00121-f001:**
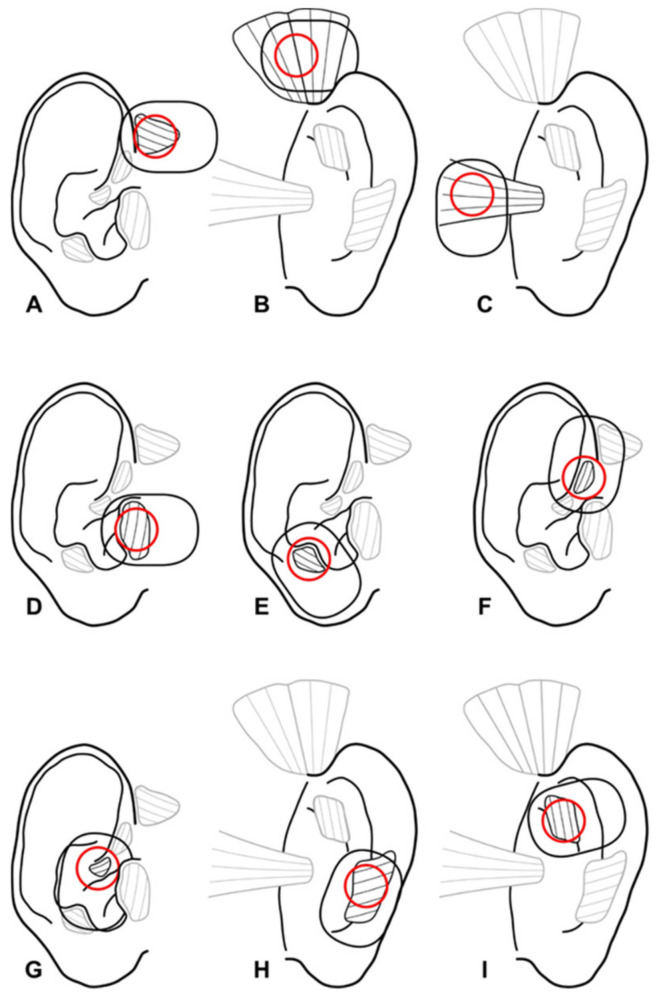
Localization of the extrinsic and intrinsic ear muscles for EMG analysis. (**A**): M. auricularis anterior. (**B**): M. auricularis superior. (**C**): M. auricularis posterior. (**D**): M. tragicus. (**E**): M. antitragicus. (**F**): M. helicis major. (**G**): M. helicis minor. (**H**): M. obliquus auriculae. (**I**): M. transversus auriculae.

**Figure 2 diagnostics-12-00121-f002:**
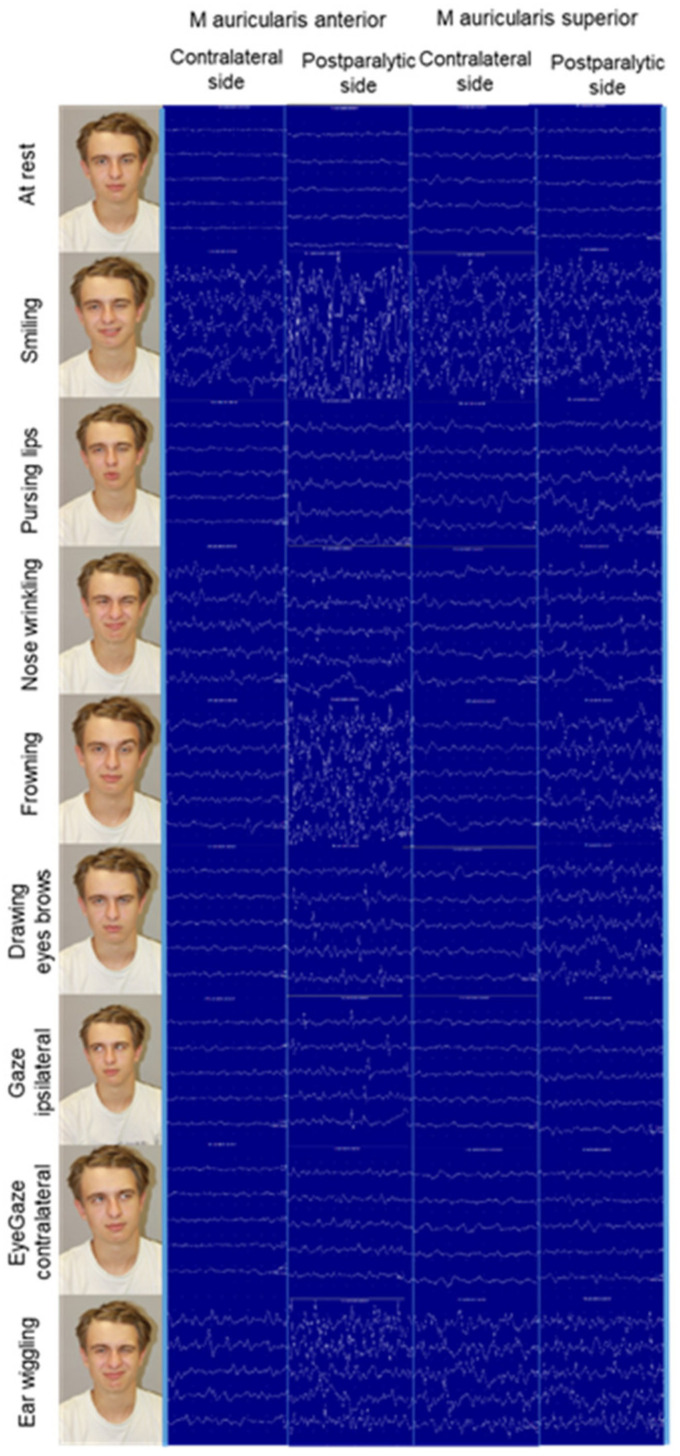
Example of a patient with postparalytic facial synkinesis on the left side. For most mimic tasks, the single-channel EMG activity was heavily increased on the postparalytic side in comparison to the contralateral side, here shown for the auricularis anterior and superior muscles.

**Table 1 diagnostics-12-00121-t001:** Healthy probands: average surface single-channel EMG activity * in the different ear muscles during mimic tasks on right and left sides.

Muscle/ Exercise	Right Side	Left Side	*p*	Muscle/ Exercise	Right Side	Left Side	*p*	Muscle/ Exercise	Right Side	Left Side	*p*
**M. auricularis ant.**	Mean ± SD	Mean ± SD		**M. tragicus**	Mean ± SD	Mean ± SD		**M. helicis minor**	Mean ± SD	Mean ± SD	
Smiling	2.91 ± 0.28	2.83 ± 0.38	0.586	Smiling	2.90 ± 0.30	3.00 ± 0	0.341	Smiling	2.77 ± 0.44	2.77 ± 0.44	1.000
Pursing lips	1.41 ± 0.79	1.66 ± 0.88	0.339	Pursing lips	1.54 ± 0.82	1.81 ± 0.60	0.277	Pursing lips	1.00 ± 1.11	1.44 ± 1.01	0.272
Nose wrinkling	0.66 ± 0.65	1.41 ± 1.08	0.069	Nose wrinkling	0.90 ± 0.87	0.80 ± 0.78	0.780	Nose wrinkling	0.11 ± 0.33	0.77 ± 1.09	0.050
Frowning	1.91 ± 1.16	1.33 ± 1.15	0.189	Frowning	1 ± 0.89	1.45 ± 1.03	0.096	Frowning	1.55 ± 1.33	1.33 ± 1.41	0.347
Drawing eyebrows	0.33 ± 0.88	0.58 ± 0.66	0.491	Drawing eyebrows	0.18 ± 0.60	0.90 ± 0.94	0.054	Drawing eyebrows	0.11 ± 0.33	0.44 ± 1.01	0.347
Ipsilateral gaze	0.41 ± 0.90	0.58 ± 0.79	0.504	Ipsilateral gaze	0.27 ± 0.90	0.18 ± 0.4	0.779	Ipsilateral gaze	0 ± 0	0.33 ± 0.70	0.195
Contralateral gaze	0	0.45 ± 0.52	0.016	Contralateral gaze	0.18 ± 0.60	0.27 ± 0.64	0.756	Contralateral gaze	0.22 ± 0.44	0.22 ± 0.44	1.000
Ear wiggling	2.75 ± 0.7	2.75 ± 0.7	NA	Ear wiggling	2.42 ± 0.78	2.42 ± 0.97	1.000	Ear wiggling	2.83 ± 0.4	2.5 ± 1.22	0.363
**M. auricularis sup.**				**M. antitragicus**				**M. trans. auriculae**			
Smiling	2.83 ± 0.38	2.83 ± 0.38	1.000	Smiling	2.91 ± 0.28	2.83 ± 0.57	0.339	Smiling	3 ± 0	2.91 ± 0.28	0.339
Pursing lips	1.16 ± 1.02	1.41 ± 0.9	0.339	Pursing lips	1.41 ± 0.79	1.33 ± 0.98	0.795	Pursing lips	1.16 ± 0.83	1.16 ± 0.71	1.000
Nose wrinkling	0.91 ± 1.08	0.66 ± 0.77	0.389	Nose wrinkling	0.58 ± 0.79	0.58 ± 0.79	1.000	Nose wrinkling	0.66 ± 0.88	1.08 ± 0.90	0.137
Frowning	2.25 ± 0.96	2.25 ± 0.96	1.000	Frowning	1.45 ± 1.21	1.54 ± 1.21	0.756	Frowning	1.66 ± 1.30	1.50 ± 1.24	0.504
Drawing eyebrows	0.50 ± 0.90	0.75 ± 0.96	0.191	Drawing eyebrows	0.33 ± 0.77	1.00 ± 1.04	0.071	Drawing eyebrows	0.50 ± 1.00	0.41 ± 0.66	0.586
Ipsilateral gaze	0.83 ± 1.19	1.00 ± 0.85	0.674	Ipsilateral gaze	0.50 ± 0.90	0.25 ± 0.45	0.339	Ipsilateral gaze	1.16 ± 1.11	1.00 ± 0.85	0.615
Contralateral gaze	0.25 ± 0.62	0.33 ± 0.65	0.723	Contralateral gaze	0.66 ± 0.77	0.33 ± 0.49	0.220	Contralateral gaze	1.25 ± 0.96	1.00 ± 0.95	0.463
Ear wiggling	2.87 ± 0.35	2.75 ± 0.46	0.351	Ear wiggling	2.85 ± 0.37	2.57 ± 1.13	0.356	Ear wiggling	2.75 ± 0.46	2.50 ± 0.92	0.170
**M. auricularis post.**				**M. helicis major**				**M. obliq. auriculae**			
Smiling	2.41 ± 0.51	2.58 ± 0.66	0.339	Smiling	3.00 ± 0	2.81 ± 0.4	0.167	Smiling	2.90 ± 0.30	3 ± 0	0.341
Pursing lips	0.91 ± 0.9	1.16 ± 0.93	0.389	Pursing lips	1.18 ± 0.98	0.9 ± 1.13	0.465	Pursing lips	1.00 ± 0.89	1.18 ± 0.98	0.506
Nose wrinkling	0.58 ± 0.79	0.75 ± 0.75	0.551	Nose wrinkling	1.18 ± 1.25	0.9 ± 0.83	0.341	Nose wrinkling	0.72 ± 1.00	0.90 ± 1.04	0.441
Frowning	1.41 ± 1.44	1.83 ± 1.46	0.210	Frowning	2.45 ± 0.93	1.9 ± 1.04	0.052	Frowning	1.90 ± 1.22	1.90 ± 1.22	1.000
Drawing eyebrows	0.50 ± 0.90	0.83 ± 1.19	0.266	Drawing eyebrows	0.63 ± 0.92	0.72 ± 1.00	0.810	Drawing eyebrows	0.90 ± 1.13	0.63 ± 0.80	0.082
Ipsilateral gaze	1.00 ± 0.95	0.83 ± 0.57	0.586	Ipsilateral gaze	0.36 ± 0.8	0.36 ± 0.5	1.000	Ipsilateral gaze	1.27 ± 0.90	1.00 ± 0.89	0.432
Contralateral gaze	0.66 ± 1.07	0.75 ± 0.75	0.809	Contralateral gaze	0.18 ± 0.4	0.18 ± 0.4	1.000	Contralateral gaze	1.18 ± 0.98	1.27 ± 1.10	0.821
Ear wiggling	2.62 ± 0.74	2.75 ± 0.46	0.351	Ear wiggling	2.86 ± 0.37	2.86 ± 0.37	NA	Ear wiggling	2.87 ± 0.35	2.50 ± 0.92	0.197

* EMG classification: 0 = no increase compared to baseline activity in resting state, 1 = slight increase, 2 = moderate increase, 3 = strong increase.

**Table 2 diagnostics-12-00121-t002:** Healthy probands: average surface multi-channel EMG activity * synchronously in several ear muscles, chewing muscles and one mimic muscle during different tasks summarized for both sided of the face.

Exercise	M. risorius	M. auricularis Posterior	M. auricularis Superior	M. tragicus	M. temporalis	M. masseter
	Mean ± SD	Mean ± SD	Mean ± SD	Mean ± SD	Mean ± SD	Mean ± SD
Smiling	2.64 + 0.50	0 + 0	2.64 + 0.50	1.91 + 1.51	2.00 + 0	1.64 + 0.50
Pursing lips	2.27 + 1.01	0 + 0	1.91 + 0.30	0.64 + 0.50	1.82 + 0.40	0.64 + 0.50
Nose wrinkling	1.09 + 0.30	0 + 0	1.36 + 0.50	0.64 + 0.50	1.00 + 0	0 + 0
Frowning	1.64 + 0.50	1.36 + 0.50	2.91 + 0.30	1.36 + 0.50	3.00 + 0	1.00 + 0
Drawing eyebrows	2.36 + 0.50	0.36 + 0.50	2 + 0	1.09 + 1.51	2.36 + 0.50	1.00 + 0
Ipsilateral gaze	1.82 + 0.98	0 + 0	1.64 + 0.50	0.36 + 0.50	1.73 + 1.01	1.09 + 1.51
Contralateral gaze	1.64 + 0.92	0 + 0	2.36 + 0.50	1.00 + 1.41	1.73 + 1.01	1.09 + 1.51
Ear wiggling	1.64 + 0.92	1.27 + 1.01	3.00 + 0	2.27 + 0.47	3.00 + 0	1.64 + 0.92
Showing teeth	3.00 + 0	0.64 + 0.50	2.27 + 0.47	3.00 + 0	2.64 + 0.50	1.36 + 0.50
Clenching teeth	3.00 + 0	0 + 0	1.36 + 0.50	1.09 + 0.30	2.64 + 0.50	2.27 + 1.01
Chewing ipsilateral	2.27 + 1.01	0 + 0	2.18 + 0.98	2.09 + 0.94	2.18 + 0.98	1.73 + 1.42
Chewing contralateral	2.55 + 0.52	0.36 + 0.50	3.00 + 0	2.27 + 1.01	2.91 + 0.3	2.27 + 1.01

* EMG classification: 0 = no increase compared to baseline activity in resting state, 1 = slight increase, 2 = moderate increase, 3 = strong increase.

**Table 3 diagnostics-12-00121-t003:** Patients with postparalytic facial synkinesis: average surface single-channel EMG activity * in the different ear muscles during mimic tasks on postparalytic and contralateral sides.

Muscle/ Exercise	Postparalytic	Contralateral	*p*	Muscle/ Exercise	Postparalytic	Contralateral	*p*	Muscle/ Exercise	Postparalytic	Contralateral	*p*
**M. auricularis ant.**	Mean ± SD	Mean ± SD		**M. tragicus**	Mean ± SD	Mean ± SD		**M. helicis minor**	Mean ± SD	Mean ± SD	
Smiling	2.71 ± 0.48	2.85 ± 0.37	0.356	Smiling	2.28 ± 0.75	2.71 ± 0.48	0.078	Smiling	2.85 ± 0.37	2.57 ± 0.53	0.172
Pursing lips	1.71 ± 1.25	1.00 ± 0.57	0.140	Pursing lips	1.57 ± 0.78	0.71 ± 0.48	**0.001**	Pursing lips	1.85 ± 1.06	0.57 ± 0.53	**0.012**
Nose wrinkling	2 ± 1	1.28 ± 1.25	0.310	Nose wrinkling	1.71 ± 0.75	1.00 ± 1.00	0.094	Nose wrinkling	2 ± 0.57	0.71 ± 0.95	**0.022**
Frowning	2.28 ± 0.75	1.85 ± 0.89	0.356	Frowning	2 ± 0.81	1.42 ± 0.97	0.280	Frowning	2.14 ± 0.69	1.28 ± 0.75	0.078
Drawing eyebrows	1.85 ± 1.21	0.42 ± 0.53	**0.016**	Drawing eyebrows	1.71 ± 0.48	0.85 ± 0.89	**0.045**	Drawing eyebrows	1.85 ± 0.89	0.14 ± 0.37	**0.007**
Ipsilateral gaze	1.28 ± 0.75	0.42 ± 0.53	0.078	Ipsilateral gaze	0.42 ± 0.53	1.00 ± 1.15	0.103	Ipsilateral gaze	1.14 ± 0.89	0.28 ± 0.48	**0.045**
Contralateral gaze	0.42 ± 0.78	0.57 ± 0.78	0.604	Contralateral gaze	0.42 ± 0.53	0.71 ± 1.11	0.604	Contralateral gaze	0.28 ± 0.48	0.42 ± 0.53	0.604
Ear wiggling	1.33 ± 1.15	2.66 ± 0.57	0.270	Ear wiggling	1.33 ± 1.52	2.33 ± 0.57	0.225	Ear wiggling	1.33 ± 1.52	2.66 ± 0.57	0.184
**M auricularis sup.**				**M. antitragicus**				**M. trans. auriculae**			
Smiling	2.57 ± 0.78	2.85 ± 0.37	0.356	Smiling	2.42 ± 0.53	2.71 ± 0.48	0.172	Smiling	2.71 ± 0.48	2.57 ± 0.53	0.356
Pursing lips	2.28 ± 1.11	1.28 ± 0.75	**0.018**	Pursing lips	1.85 ± 1.06	0.57 ± 0.78	0.035	Pursing lips	1.85 ± 0.69	1.00 ± 0.81	0.078
Nose wrinkling	2.28 ± 0.75	1.00 ± 1.15	0.093	Nose wrinkling	2.28 ± 0.75	0.57 ± 1.13	**0.011**	Nose wrinkling	2.42 ± 0.53	1.14 ± 1.06	0.063
Frowning	2.28 ± 0.75	1.85 ± 1.06	0.510	Frowning	2 ± 0.81	1.14 ± 0.37	**0.017**	Frowning	1.71 ± 0.95	1.42 ± 0.53	0.522
Drawing eyebrows	2.28 ± 1.11	0.42 ± 0.53	**0.004**	Drawing eyebrows	1.28 ± 1.25	0.28 ± 0.75	**0.038**	Drawing eyebrows	1.71 ± 0.75	0.71 ± 0.95	0.062
Ipsilateral gaze	1.28 ± 1.11	1.00 ± 1.00	0.457	Ipsilateral gaze	1 ± 0.57	0.57 ± 0.53	0.078	Ipsilateral gaze	0.85 ± 0.69	1.28 ± 0.95	0.289
Contralateral gaze	0.85 ± 1.21	0.71 ± 0.75	0.766	Contralateral gaze	0.42 ± 0.53	0.42 ± 0.78	1.000	Contralateral gaze	0.57 ± 0.53	1.42 ± 1.13	0.111
Ear wiggling	2 ± 1	2.33 ± 0.57	0.742	Ear wiggling	1 ± 1.73	2.00 ± 1.00	0.225	Ear wiggling	1.33 ± 1.52	2.00 ± 1.00	0.667
**M. auricularis post.**				**M. helicis major**				**M. obliq. auriculae**			
Smiling	2.71 ± 0.48	2.28 ± 0.48	0.200	Smiling	3 ± 0	3.00 ± 0	NA	Smiling	2.71 ± 0.48	2.57 ± 0.53	0.356
Pursing lips	1.85 ± 0.89	0.71 ± 0.75	**0.005**	Pursing lips	1.71 ± 1.11	0.57 ± 0.53	**0.047**	Pursing lips	1.85 ± 1.06	0.71 ± 0.95	0.103
Nose wrinkling	2.57 ± 0.53	1.14 ± 1.21	**0.008**	Nose wrinkling	2.42 ± 0.53	1.42 ± 1.13	0.086	Nose wrinkling	2.28 ± 0.48	1.14 ± 1.21	0.066
Frowning	2.14 ± 0.69	1.57 ± 1.27	0.413	Frowning	2.57 ± 0.53	1.71 ± 0.95	**0.017**	Frowning	2 ± 1	1.85 ± 0.89	0.604
Drawing eyebrows	2 ± 1.15	0.28 ± 0.48	**0.007**	Drawing eyebrows	2.14 ± 0.69	0.28 ± 0.75	**0.0001**	Drawing eyebrows	1.71 ± 0.75	0.43 ± 0.54	**0.022**
Ipsilateral gaze	1.14 ± 1.06	1.00 ± 0.81	0.736	Ipsilateral gaze	0.85 ± 0.69	0.28 ± 0.48	**0.030**	Ipsilateral gaze	0.71 ± 0.48	1.28 ± 1.11	0.231
Contralateral gaze	0.85 ± 0.69	1.00 ± 0.81	0.604	Contralateral gaze	0.42 ± 0.53	0.42 ± 0.53	1.000	Contralateral gaze	0.71 ± 0.48	1.71 ± 1.25	0.111
Ear wiggling	2 ± 1	2.66 ± 0.57	0.423	Ear wiggling	1.33 ± 1.52	2.33 ± 1.15	0.225	Ear wiggling	1.33 ± 1.52	2.66 ± 0.57	0.184

* EMG classification: 0 = no increase compared to baseline activity in resting state, 1 = slight increase, 2 = moderate increase, 3 = strong increase; significant values (*p* < 0.05) in bold.

## Data Availability

The datasets used during the current study are available from the corresponding author upon request.

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
