# Peer review of "Electromyography of Extrinsic and Intrinsic Ear Muscles in Healthy Probands and Patients with Unilateral Postparalytic Facial Synkinesis"

_diagnostics, 2022, doi:10.3390/diagnostics12010121_

Round 1

Reviewer 1 Report

This is a well written paper describing EMG recordings from the intrinsic and extrinsic muscles of the ear. A couple of minor suggestions for the paper. The cadaver study is not mentioned in the introduction and the context or progression of the study is not described (i.e., first we did cadavers to look at muscle localization, then we did health controls, then patients). Perhaps describing the cadaver study in the introduction would help with this as opposed to lumping it into the methods with no context.

In the methods section, there is no mention that surface electrodes were adapted (cut and/or sized) to the muscles being measured. The supplemental file clearly shows the surface electrodes with different sizes but this is not mentioned. This would help with reproduction of the results for others.

There is confusion on the metric used to evaluate the EMG responses. The discussion briefly cites a paper where the qualitative use of the 0-3 scale comes from but the paper does not mention any of this before hand. Why are the values in Table 1 so granular when the outcomes are only 0-3 (not 2.3, 1.8, etc.)? This is in contrast to Table 2 and Table S1. Some more explanation on the methodology used to get the numbers is appropriate. A table description would also be helpful. Importantly, how do you determine the difference between a slight and a moderate increase in EMG? Is this a purely subjective evaluation or is some quantitative threshold used?

Overall, the paper should be accepted once these corrections are performed and clarity is brought to the issues outlined.

Reviewer 2 Report

The authors have submitted an elegantly performed study that evaluates co-activation of the ear muscles when smiling, pursing the lips, wrinkling of the nose and other facial movements. The authors review previous literature, noting that EMG data on human ear muscles are sparse. Based on anatomic preparations of all ear muscles on two cadavers, the authors performed EMG examination in 12 healthy participants and seven patients with unilateral post paralytic facial synkinesis. Although a relatively small sample size, this study provides an important information concerning synchronous activation of ear muscles during different facial expression. The study is well written and referenced and supported by helpful figures and Tables. I appreciate the amount of work and effort you put into this article. Overall, I feel the manuscript would be of interest to the diagnostics readership.
